# Snacks and The City: Unexpected Low Sales of an Easy-Access, Tasty, and Healthy Snack at an Urban Snacking Hotspot

**DOI:** 10.3390/ijerph17207538

**Published:** 2020-10-16

**Authors:** Caroline Schlinkert, Marleen Gillebaart, Jeroen Benjamins, Maartje P. Poelman, Denise de Ridder

**Affiliations:** 1Department of Social, Health and Organizational Psychology, Utrecht University, 3584 CS Utrecht, The Netherlands; m.gillebaart@uu.nl (M.G.); j.s.benjamins@uu.nl (J.B.); d.t.d.deridder@uu.nl (D.d.R.); 2Department of Experimental Psychology, Helmholtz Institute, Utrecht University, 3584 CS Utrecht, The Netherlands; 3Chair Group Consumption and Healthy Lifestyles, Wageningen University & Research, 6708 PB Wageningen, The Netherlands; maartje.poelman@wur.nl

**Keywords:** nutrition education, public health, urban field experiment, nudging intervention, food marketing

## Abstract

While many people declare an intention to eat and snack more healthily, a large body of research has found that these intentions often do not translate into actual behavior. This failure to fulfil intentions is regularly attributed to the obesogenic environment, on which basis it is assumed that changing the food environment may lead to more healthy snacking behavior. To test this premise in real life practice, the present research project investigated whether making a healthy snack easy-to-access in an urban environment characterized by unhealthy snacking would support people in their intentions of purchasing more healthy snacks. The urban snack project consisted of three phases. In Phase 1, a hotspot location for unhealthy snacking was determined by using a Global Positioning System to track people’s snacking locations and a survey to verify the location. In Phase 2, an attractive snack was developed that met consumers’ criteria of what constituted a healthy and tasty snack, together with corresponding branding that also included a small food truck from which to sell the newly developed snacks. In Phase 3, the snack was sold from the food truck located at the previously determined unhealthy snacking hotspot. We counted the number of snacks sold and canvassed people’s opinions about the snack and its branding, finding that in spite of people’s appreciation for the snack, the food truck, and the branding, actual sales of the snack were very low. In the Discussion, we name predominant eating and purchasing habits as possible reasons for these low sales. Future research could focus on placing the healthy snack directly beside people’s habitual snack purchase location to ensure that the new choice gets better recognized. Overall, the findings suggest that merely placing healthy snacks more prominently in people’s food environment is not sufficient to lead people to snack more healthily.

## 1. Snacks and The City: Unexpected Low Sales of an Easy-Access, Tasty, and Healthy Snack at an Urban Snacking Hotspot

Snacking has been increasing rapidly for decades, with the number of people who consume three or more snacks a day rising fourfold between the 1970s and 2005 [1]. Recent estimates indicate that some 25–35% of daily energy intake now comes from snacking [2,3]. This is problematic because snacking is strongly associated with unhealthy diets [4,5] mainly because snacking tends to involve the consumption of energy-dense foods that are rich in sugar and/or saturated fat and are of poor nutritional quality [6]. The inherent health risks of snacking, combined with its increasing prevalence, has led to numerous health education interventions aimed at decreasing unhealthy snacking behavior. However, these interventions have mostly proven unsuccessful [7,8].

The failure of such interventions to change snacking behaviour may not be directly attributable to lack of public awareness of food health issues, however, since people are generally able to identify the key elements of a healthy diet. For instance, a systematic review of research on various aspects of a healthy diet found that people typically cite the need for balance, variety, fruit and vegetables, and fresh produce as elements of a healthy diet [7], though the same review also revealed that people find the various guidelines for a healthy diet to be complex and confusing. Moreover, the gap between people’s food knowledge and healthy eating intentions and their actual (often unhealthy) eating behavior has been shown to be very wide [9]. This gap becomes especially clear from research showing that the majority of people do not meet recommendations for a healthy diet [7].

The urban food environment is increasingly recognized as a major factor affecting the outcome of people’s efforts to maintain a healthy diet [10]. The “obesogenic” food environment, characterized by the omnipresence and easy accessibility of palatable but energy-dense and processed foods, has been shown to contribute significantly to high levels of snack consumption [11]. Accordingly, the food environment should be a key component of any intervention that aims to target and change snack behavior [10]. In addition to the physical presence of unhealthy foods, the current food environment also communicates the social norm message that consuming unhealthy foods is ”appropriate” [12,13]. For these reasons, even when people have sufficient diet information, food knowledge, and healthy eating intentions, the current food environment makes it extremely challenging for them to change their snacking behavior [14]. Addressing this environmental challenge, the current research project examined in a real-life setting whether making healthy snacks easy-to-access in a city hotspot location for unhealthy snacking would lead people to make healthy snack choices. With this we aim to contribute to a more comprehensive understanding of the relation between snack choices and factors contributing to obesity, thereby informing the development of more efficient health measures [15]. Our research aim thus supports the long-standing health promotion goals mentioned by the World Health Organization (https://www.who.int/healthpromotion/conferences/previous/ottawa/en/).

## 2. Problems Arising From The Food Environment

In seeking to better understand the mechanisms of the food environment that lead people to unhealthy snacking, researchers have examined the proximity impact of unhealthy snacks. The relevance of such research is underlined by studies that have demonstrated an association between the increased density of fast-food outlets and increased incidences of cardiovascular diseases and unhealthy lifestyles [16]. Experimental lab studies have indeed confirmed that the proximity of unhealthy snacks, typically within arm’s reach, increases the likelihood of unhealthy snack consumption [17,18,19]. However, studies have also found that placing *healthy* snacks in greater proximity increases the consumption of these healthier products. Specifically, field studies conducted at unhealthy snack hotspots (such as travel locations) have shown that placing healthy snacks near the cash register significantly increased their sales, often without customers showing awareness of the cause when subsequently asked about their purchases [20,21]. These observations demonstrate that the proximity and easy availability of snacks play an important role in the purchase process. Such findings are also consistent with the idea that the majority of snack purchases are made impulsively or as part of habitual routine behavior without much reflection on which product is bought or the reasons for the purchase [20].

Given these findings, it is reasonable to assume that an effective way to intervene in the food environment would be to offer healthy snack alternatives in proximity to unhealthy options. One benefit of such proximity is that the alternative healthy snacks would thus be offered in peoples’ habitual snack purchase environments, thereby increasing the likelihood of counteracting people’s routine behavior of buying unhealthy snacks [20]. Another advantage of such placement is that the prevailing associations that consumers already have with existing unhealthy snacks might in this way be ”hijacked” by the new and healthier alternative [22]. Whether people have healthy snacking intentions or not, such placement would confront them with a healthy choice in their personal environment, potentially increasing the purchase and consumption of healthier snacks [23].

Associated with the problem of the proximity and availability of unhealthy food in people’s food environments is the fact that unhealthy snacking tends to take place at out-of-home locations where people are exposed to numerous outlets offering unhealthy snacks [24,25]. Everyday observations alone confirm that travel locations such as highway service stations, train stations, and airports have an especially high density of unhealthy snack outlets. Indeed, certain definitions of snack behavior even specifically include an ”on the go” component, highlighting ”portable food items such as fruit, crisps, yogurt, or sandwiches that can be eaten on the move and are clearly associated with eating in the car, on the train, or at a desk” [26]. Snack outlets have become increasingly popular over the past few decades, especially among younger people [27,28], while studies have found that foods eaten outside the home contribute to disproportionately high levels of fat intake [29,30]. As such, the popularity of snack outlets is thought to be one of the key drivers of increasing levels of being overweight and obesity [29].

Nevertheless, most snacking seems to take place at home or otherwise indoors such as at work or school [31,32]. For example, a study of Norwegian adults [32] found that 81% of all snacks were consumed either at home (58%) or at work/school (23%). Similar results have been found in the Netherlands [33,34]. In the case of snacking at home, however, people at least have the option of not bringing unhealthy snacks from the supermarket home in the first place and thereby avoid being tempted by unhealthy snacks in their kitchen cupboards [35]. By contrast, when snacking outside the home and “on the go”, there seem to be hotspots for unhealthy snacking that offer only few and uninspiring healthy snack alternatives [36]. Several studies have accordingly suggested that healthy food choices should be made at least as readily available outside the home as unhealthy foods, especially with a view to improving younger people’s diets [37].

However, merely making fruits and vegetables more accessible has thus far not proven able to solve the problem of unhealthy snacking. Field interventions on healthy (snack) eating have generally been shown to be more effective in reducing unhealthy eating than in increasing healthy eating or reducing total eating [38]. Similarly, interventions aimed at increasing fruit and vegetable choice have only shown moderate effects [23]. Moreover, people generally seem to have a preference for unhealthy snacks (whether sweet and sugary or fatty and salty) over healthy snacks (wholegrain, no added sugars and/or salt, etc.), as evidenced by research on the “unhealthy equals tasty” intuition [39], including a series of studies using a US sample that showed that people tend to infer food items are tastier when they are less healthy and that they specifically prefer unhealthy snacks when the hedonic goal of eating enjoyment is uppermost [39,40]. Consistent with these observations is the finding that people experience less pleasure from consumption when they focus on health facts about food [41]. In order to support people’s intentions to eat healthily, there is, therefore, clearly a pressing need to offer healthy “on the go” snacks that appeal to taste as much as unhealthy snacks do [42,43].

## 3. The Project Objectives

This research project explores the potential effectiveness of changing environmental factors to support people in their intentions of snacking healthily, including factors related to proximity, on-the-go snacking, and taste appeal. The general aim of the project was to make healthy snacking easy-to-access and appealing for people when they are “on the go” by selling an attractive and healthy snack with a corresponding branding at an urban hotspot for unhealthy snacking. We expected that people would like the snack and the branding and therefore buy the product, potentially even switching to the healthy snack in place of their regularly purchased unhealthy snacks. To realize this objective, the project involved scientific experts from multiple disciplines including urban geography, psychology, medicine, and public health, as well as stakeholders of the local government and leading food marketeers.

## 4. Methods

### 4.1. Participants

For each of the studies conducted, people were randomly approached on the streets in community and public spaces in the target city of Utrecht in the Netherlands. Ethical approval for the project was obtained from the ethical review board at the Faculty of Social Science of Utrecht University [Approval number: FETC17-087]. Although everyone between 16 years and older could participate, respondents with low socioeconomic status were explicitly encouraged to take part in all phases of the project. People from disadvantaged groups in the Netherlands are more often overweight (61% overweight) compared to those with a higher socio-economic status (41% overweight; [33] in spite of their intentions to live a healthier life [44]. Similar trends apply in other industrialized countries [45]. Moreover, socially disadvantaged groups are generally underrepresented in research studies due in part to their low response rate in surveys [46]. Including this population in research is essential for learning more about their preferences and behaviors in order to develop effective health measures targeted at this group.

### 4.2. Procedures

The urban snack project consisted of three phases, the first of which was to identify the best location for selling the healthy snack. This was accomplished by tracking people via a Geographic Positioning System (GPS) on a mobile application, followed by a field survey of opinions about the snack offers at this location in order to verify it as a hotspot for unhealthy snacking. The second phase consisted of a concept-mapping study and a tasting session to find out more precisely what people want in their ideal snack in order to develop a new snack with only healthy ingredients that would meet the “tasty” criteria. The development of this snack was also based on information previously gathered about people’s ideas of an ideal snack and how that ideal relates to their associations with healthy and unhealthy snacks [43]. The third phase consisted of a field study in which the newly developed snack was sold from a small food truck located in the previously detected snack hotspot. In this field study, we aimed to examine whether offering the newly developed healthy and attractive snack with a correspondingly branded food truck would make people buy and eat the snack. The branding was added to give the snack a professional look, since the appearance of food is an important factor in marketing [47]. The effects of our intervention were measured by the volume of snack sales and by responses to questions about how much people liked the snack and the food truck branding. Given the setup of this project, the detailed methods of each study will be presented together with the results.

## 5. Results

### 5.1. Phase 1. Detecting the Snack Hotspot Location

Phase 1 of the urban snack project aimed to identify the main places where people in the target city primarily snack unhealthily when they are “on the go”. This aim was based on our assumption that identifying a hotspot location for unhealthy “on the go” snacking would provide us with the best opportunity for intervening in an obesogenic environment to help people choose a healthier snack when not at home. This first phase consisted of two field studies: a GPS study to detect the snack hotspot location and a subsequent field survey to verify this location based on people’s opinions about the snack offers available at this spot.

***GPS study method***. Participants were followed for three days with a snack diary mobile application (app). This app aimed to record GPS coordinates along with self-reports to identify the main places where people snacked in the city and thus to identify the optimum hotspot for selling our healthy snack. The participants kept track of their food intake by logging every instance of food intake. This logging consisted of starting the app, turning on the GPS function, and then answering a question about which particular food was logged and whether it was breakfast, lunch, dinner, or a snack. Eating snacks were ultimately analysed. The GPS data were stored when a GPS connection could be made and the participants were directed to turn off the GPS after logging their food intake in order to minimize the app’s usage of their mobiles’ battery life.

GPS data were stored in a separate file for each participant per day over three days, enabling us to collect 114 files containing 4042 raw GPS coordinates of 45 participants. However, the GPS data of 7 participants could not be used in some cases because no coordinates had been logged between switching on and off the GPS and in one case, because only coordinates outside of Utrecht were logged as the participant appeared to live in a Dutch city close by. Consequently, the GPS data files of a total of 38 participants were included: (M_age_ = 30 years (SD = 8.2); 79% women; M_BMI_ = 23 (SD = 3.0)).

***GPS raw data analysis.*** We plotted all of the usable raw GPS data points (latitude and longitude) on a Google map using hamstermap.com and then inspected the data for clusters. There were clusters in the target neighborhood of 300 m wide centered around different residential addresses (see the map on the left in Figure 1). However, in all these clusters, only the data of one participant were present. A similarly large cluster was also present at the local train station, however, and this cluster contained the data of four participants (see the map on the right in Figure 1).

***Analysis of GPS instances.*** The raw GPS data were condensed to median latitude and longitude data for each instance that food intake was logged. A single instance of GPS logging was defined as the period between the moment the GPS was switched on and the moment it was switched off. The frequency of these instances (i.e., the average number of GPS logs per day and the average number of days the participants logged) and the locations of these instances were then further examined. On average, people logged GPS for 2.7 days (Range 1–4, SD = 1.2) at an average rate of 3.06 times per day (Range 0.3–11, SD = 2.5). A total of 352 GPS instances were available in the data, of which 265 instances were within the city limits of Utrecht. Of the participants with GPS logs in the train station area, on average, 21.7% of all their GPS log instances were in the station area (Range 6.7%–42.1%, SD = 14.8%). As with the raw GPS analysis, plotting the median latitude and longitude also showed clusters of only single participants centered around residential addresses and a cluster in the station area containing four participants.

Overall, the results of the GPS analyses revealed that snacking took place largely when people were at home (see the map on the left in Figure 1). When snacking occurred “on the go”, it was the local city train station area that turned out to be the place where people snacked most often (see the map on the right in Figure 1). Anecdotal feedback from the participants revealed that one important reason for snacking at home was that they considered out-of-home snacking to be expensive and thus preferred to buy snacks from local supermarkets to eat at home. For people who snacked “on the go” (e.g., at the train station), this occurred at locations with a higher density of fast food and takeaway outlets and in places where they did not have the opportunity to snack at home.

***Snack hotspot field survey.*** We conducted a second field study to confirm whether the local train station area was indeed a hotspot for unhealthy snacking. Train travelers (N = 161; M_age_ = 32 years (SD = 17.1); 56% women; M_BMI_ = 23 (SD = 3.1)) were asked what they bought most often at the station and what they thought about the snack offers at this location. Overall, travelers most often bought snacks at the station (35%), followed by drinks (20%), lunch (18%), dinner (14%), and breakfast (13%). Most travelers (56%) indicated that they typically bought a snack between 12:00 and 19:00. Further, 44% of the respondents said they found unhealthy snacks to be very easily accessible at the train station, while 23% found healthy snacks rather difficult to find at the station. In addition, nearly half (47%) of the respondents said that they would appreciate a wider range of healthy snacks in the station area.

***Summary Phase 1.*** The GPS study revealed that most people snack at home, which is in line with findings from earlier research [32]. However, when people were snacking “on the go”, the local city train station area seemed to be a hotspot for unhealthy snacking, which again is consistent with previous research that has identified “on the go” locations as a particular source of unhealthy food intake [36]. Indeed, those train travellers who reported most often buying snacks at the station said that healthy snacks were difficult to find and declared that they would appreciate a wider range of healthy snacks at this location. The station area thus seemed to be a good place to offer a healthy and attractive snack as an alternative “on the go” snack.

### 5.2. Phase 2. Developing an Ideal Snack With Healthy Ingredients That Met The “Tasty” Criteria

Phase 2 of the urban snack project aimed to develop a new attractive snack to sell at the unhealthy snack hotspot. We assumed that if people were offered a tasty and filling snack, they would be more likely to buy the snack “even though” the snack was made of healthy ingredients. We therefore investigated people’s ideas and associations regarding their ideal snack [43]. In this way, we were better able to understand what people considered to be an attractive snack without unduly influencing their answers or evoking negative associations and presumptions about the (un)healthiness of snacks. This second phase consisted of a concept mapping study, a pilot tasting session of a first iteration of the new snack, and several brainstorming sessions with our research team and food designers of a leading marketing company in the Netherlands who specialized in healthy food marketing campaigns (www.foodcabinet.nl) to agree upon the final version of the new snack.

***Concept mapping to identify the ideal features of a snack.*** We conducted a concept mapping study to map out which features people associated with their “ideal snack” [43]. Concept mapping is a brainstorming technique that can be used to collect the ideas of a group of people from different backgrounds and to describe their ideas in a graphical form [48]. We held several small group meetings in our target neighborhood (each with 5 to 20 people), giving the participants cards (N = 72 cards) that each had a snack feature written on them, such as “tasty”, “vitamins”, “price”, “handmade”, “filling”, “light”, “crunchy”, “cold”, etc. In a first step, the participants were asked to rate each snack feature according to how important they thought it was for this feature to be present in their ideal snack, using a 7-point Likert scale ranging from 1 = “not at all” to 7 = “very much”. In a second step, participants were asked to stack the cards by categories according to how they thought the features belonged together contextually. The participants were not allowed to put all their cards on one stack or to make a single stack for each card. In addition, no card was allowed to be used more than once [49].

Based on these two rating procedures, we conducted a concept mapping analysis to ascertain which features belonged together and how much importance people attached to each feature (see the R-CMap software package in [50]). The data of 55 participants were included in the final data set (60% women; M_age_ = 34 years [SD = 17.1]; 27% lower education; 38% higher education; 18% university education; and 16% high school only). Figure 2 presents the final five clusters and feature ratings.

Figure 2 lists only those features rated 4 or higher, since features with a lower rating implied insufficient agreement on the importance of this feature among the participants [50]. Clusters 1–4 involve sensory and health-related characteristics, whereas Cluster 5 concerns price-related features. The five highest rated features are “tasty”, “fresh”, “healthy”, “vitamins”, and “nutritious”, all of which belong to Cluster 3. In our earlier study on people’s associations with an ideal snack, we also found that participants (N = 1087) rated the feature “healthy” among the top five, alongside features related to sensory characteristics such as “warm”, “sweet”, “cold”, and “savory” [43]. It thus seems that people have idiosyncratic ideas about the sensory characteristics of their ideal snack, though “healthy” appears to be a common feature, and we tried to take these different tastes into account when developing our healthy version of an ideal snack. Since “price” was also rated as important, forming its own cluster of features, we also made sure that the snack price was appropriate to the selling location (see Phase 3).

***Pilot tasting session for the first versions of the ideal snack.*** A pilot tasting session was conducted in the target neighborhood for the first two versions of the ideal snack. The snack was created by food designers on the following basis: the results of the association study regarding an ideal snack [43]; the findings of the ideal snack concept-mapping study; and the conclusions from brainstorming sessions held with our research team and the food designers about the interpretation of the data and possible snack options. The first versions of the two snacks for piloting were a vegetable wrap and an apple and raisin pie with a wholegrain pie crust, neither of which contained added sugar or salt. The snacks are depicted in Figure 3.

For the pilot study, people were randomly approached on the street for a “fast snack-tasting” test and we therefore did not gather demographical information. We randomly approached 30 members of the public in a community space and asked them to try one of the two snacks and to rate them according to their appearance, taste, and healthiness, as well as to indicate whether they would actually buy the snack and how much they would pay (using Likert scales ranging from 1 “not at all” to 5 “very much”). For the apple and raisin pie snack, people on average liked how it looked (M = 3.6; SD = 0.7) and enjoyed the taste (M = 3.7; SD = 0.8), further indicating that the snack looked quite healthy (M = 4.2; SD = 0.8) and that they would probably buy it (M = 3.5; SD = 0.8) for a price of about 1 euro. Similar results were found for the vegetable wrap snack. People on average liked how it looked (M = 3.8; SD = 1.0) and enjoyed the taste (M = 3.9; SD = 0.9), thought it looked quite healthy (M = 3.9; SD = 0.9), and said they would probably buy it (M = 3.7; SD = 1.2) for a price of 1.50 euro. However, the participants’ verbal reports also indicated that the vegetable wrap was not perceived as practical for eating on the go and consequently we decided a pie would be better as an “on the go” snack on account of being more convenient to eat while walking. In our brainstorming session with the food designers, we further agreed that pies would be easier to vary and to brand since the public are very familiar with pies.

Therefore, for the final version of the snack we created a savory vegetable version of a pie snack that resembled the apple and raisin pie. Please see Figure 4a for a picture of both versions. The two final pie snacks were thus the apple and raisin pie and a pie with a vegetable (ratatouille) filling without any added sugar or salt in the filling. The pie crust was made from wholegrain wheat flour, egg, plant-based butter, water, and a pinch of salt. With these two pies, we were able to offer people the choice between a savory and a sweet snack, thereby catering to the individualistic snack preferences indicated in the concept mapping study and the “ideal snack” association study [43]. Appendix A presents the full ingredients of the pies. Together with the food designers, we then created the brand “Van Taartjes” (“Little Pies”) for the field experiment, with a correspondingly styled menu board and clothing for the sales staff. Please see Figure 4b,c for pictures of the food truck and the corresponding branding.

### 5.3. Phase 3: Selling The Healthy Snack at an Easy-to-Access Location In the Unhealthy Snacking Hotspot

The third phase of the urban snack project consisted of a field intervention study in which the newly developed snack was sold from a small food truck located at the previously determined snack hotspot. This field study aimed to examine whether offering the new tasty and healthy snack with an attractively branded food truck would make people choose healthier snacks when “on the go”.

***Food truck intervention study methods.*** The food truck was placed in the train station area of the target city, i.e., in the snack hotspot determined in Phase 1, for three months. Please see Figure 5 for a detailed description of the procedures followed on each selling day. In Phase 1, the majority of people had indicated that they typically bought something at the station between noon and seven o’clock (57%) and we therefore focused our selling days and hours around the peak periods of the afternoon rush hour, i.e., between 2:30 and 6:00 pm during the working week. In order to minimize competition from the unhealthy snacks on sale in the station, the food truck was placed in such a way that travellers had to pass it either on entering or leaving the station where they would have many opportunities to purchase unhealthy snacks.

***Opinions about the location, the appearance of the food truck, and the “Little Pies” snacks.*** Halfway through the experiment, we began asking people passing by the truck to try (one of) the two snacks and to rate the selling location, the design of the truck, and the snack through a face-to-face interview survey. All questions were answered on Likert scales ranging from 1 “completely disagree” to 7 “completely agree” and the total sample consisted of 50 participants (56%women; M_age_ = 30.4 years [SD = 17.0]). This survey revealed that participants liked the location (*M* = 5.5; *SD* = 1.7), liked the design of the food truck (*M* = 6.2; *SD* = 1.0), found that the snacks looked tasty (*M* = 5.5; *SD* = 1.1) and looked healthy (*M* = 5.7; *SD* = 1.3), enjoyed the taste (*M* = 5.9; *SD* = 0.9), and would potentially buy a snack from the food truck next time they passed by (*M* = 4.4; *SD* = 1.9). However, they found the price rather high (*M* = 5.2; *SD* = 1.2) and we therefore lowered the price from €2.50 (corresponding to the average price of the most favoured snacks bought at the train station) to €1.75.

***Opinions about the taste of the “Little Pies” snack.*** We gathered contact information both from our actual customers and from the people passing by our food truck throughout the entire three month selling period. At the end of the selling period, we sent each contact an email with an online questionnaire including questions asking them to recall and review the design of the food truck, whether they had bought our snack, and if so, how much they had enjoyed it. The total number of respondents comprising our final sample was 75, of whom 61 completed the survey (79% women; M_age_ = 29.2 years [SD = 13.2]). The participants, on average, cared about healthy eating (*M* = 5.7; SD = 0.9). The results revealed that 56% of the participants had noticed the food truck at least once a week and that 23% had bought at least one snack from the truck on at least one occasion. People on average reported that they had quite liked the snack (*M* = 4.6; SD = 1.4; Likert scale ranging from 1 “very bad” to 7 “delicious”), had found the price acceptable (*M* = 3.4; SD = 1.7; Likert scale ranging from 1 “very cheap” to 7 “very expensive”), and had found the food truck to be fairly appropriate for the selling location (*M* = 4.4; SD = 1.6; Likert scale ranging from 1 “not at all” to 7 ‘”very much”).

***Snack sales.*** In spite of this positive feedback and in spite of the large number of potential customers that passed by the truck every day (amounting to several thousands), the sales data revealed that snack sales were very low, with an average of only five snacks purchased per day during the entire three-month selling period (N = 48 selling days; M = 5.5 sold snacks per day [SD = 3.2]). In our efforts to boost sales, we made various additional arrangements during the sales period, including the previously mentioned price reduction. However, none of these adjustments led to higher sales. For example, after initially offering our snack without explicitly advertising it as a healthy product, we later openly promoted this feature by placing a “healthy snacks” board on the food truck to give potential customers an additional reason to buy the snack; but again, this did not lead to any increase in sales. Other marketing tactics included offering a free snack to taste and a social proof experiment in which a queue (consisting of research assistants) was formed in front of the food truck to stimulate interest, but again, neither of these tactics led to increased sales.

## 6. General Discussion

The urban snack project arose from the thesis that the obesogenic environment largely dictates our food choices. The omnipresence of unhealthy food and the lack of healthy alternatives pressures people into purchasing unhealthy snacks, especially in the snack hotspots often found in urban areas [51,52]. Our project aimed to facilitate people’s intentions to eat more healthily by giving the public the option to choose a healthy snack in one such hotspot by offering a healthy and tasty snack prominently marketed in an optimum location. By first examining and identifying people’s unhealthy snacking hotspots, we found that people mostly snack at home. We also found, however, that when snacking occurred “on the go”, the local city train station area was a major hotspot for unhealthy snacking and one that offered no substantial alternative options for healthy snacking (Phase 1). Second, we developed a new attractive snack that both met people’s criteria for tastiness and was also made from healthy ingredients. This snack was based on people’s ideas about their ideal snack, including various sensory characteristics such as “tasty”, “fresh’”, and “savory”, as well as the attribute of being “healthy”, which was also among the five features most frequently cited as important by the participants in our research [43]. Despite our evidence-based assumptions and methodological rigor, the very low sales of our healthy snacks revealed that our efforts to make healthy snacking easy-to-access and more attractive were insufficient to lead a significant number of people to choose our healthy snack (Phase 3). In the best case interpretation, our intervention might have made people more aware of their unhealthy snacking patterns and as such, could have served as a starting point for them to change their snacking behavior. However, our results imply that additional measures are needed beyond ease of access and branding, etc., to facilitate people’s intentions to snack more healthily.

One possible explanation for our findings could be that people found the snack “too healthy”. Indeed, studies have shown that health motives play a role in food choices [53] and that healthy foods are specifically less liked when they are presented as a morally better choice since food moralization can interfere with pleasure in eating [54,55]. We doubt that this was the case here, however, since the train travelers passing by our food truck thought the snack looked healthy but also (fairly) tasty and our actual customers appreciated the pie’s taste (Phase 2 and Phase 3). Moreover, the respondents consistently expressed a desire for a wider variety of specifically healthy snacks to be on sale at the train station (Phase 1).

A more plausible explanation for the results, in our view, relates to the force of habit and the multiple influences of the food environment on snacking intentions and behavior. Many of our respondents indicated that they would like to snack more healthily. The train travellers specifically stated a preference for a larger range of healthy snacks at the train station (the snack hotspot identified in Phase 1) and the participants in the “ideal snack” association study frequently used the term “healthy” when describing their ideal snack (Phase 2 concept mapping results; and [43]). However, even when people have healthy eating goals, their intentions may be crowded out by their preoccupations with other immediate daily life goals such as hurrying to catch trains, preparing for important meetings, or enjoying a day out. These competing goals may lead people to “forget” or subsume their healthy eating goals when going about their daily snacking routines, thus reducing the likelihood of any behavioral switch towards a healthier choice of food [56,57,58]. Moreover, the obesogenic environment consists of more than just unhealthy offerings: the impact of the obesogenic environment also exploits and relies upon habitual responses to this environment, i.e., people’s habits of buying unhealthy food at certain times and places. Our findings suggest that the negative health impacts of this environment cannot be countered or overcome easily, as, for example, in our case, by “simply” altering the supply of healthy food in the environment and offering it prominently as an alternative to unhealthy snacks.

### Study Limitations and Future Directions for Offering Easy-to-Access Healthy Snacks

Our project had certain limitations. First, it could be the case that our snack was not optimally positioned so as to be considered an appealing snack alternative by travellers. The food truck was placed directly by the entrance of the station’s main hall so that every person who wanted to enter the train station had to pass by the truck before being confronted with the usual snack outlets. We assumed this to be a good location because several thousands of people a day had to pass by our truck. The truck was thus clearly visible to everyone entering or leaving the train station. Moreover, food trucks had used this spot before to sell all kinds of snacks and drinks. Nevertheless, it could be that the choice to purchase a snack “on the go” is only made by travellers *after* they have entered the train station hall and when they already have a train ticket and know how much time they have left before their train leaves. Conversely, many travellers arriving on trains at the station may immediately consume snacks while still in the station hall before even seeing our food car. Further increasing the proximity to people’s routine snack choice location, preferably directly beside a similar, popular unhealthy snack item in a familiar snack store, could therefore increase sales in a future project [59,60]. Another option could be to more prominently promote the snack as healthy in the form of a health campaign, maybe even in collaboration with other parties such as the local government or popular snack outlets. This looks to bundle forces by not only offering a new snack, but also educating people about healthy snack options. Second, the offer available from the food truck could have been too limited, consisting solely of our two new snacks, meaning the only choice customers had was between a sweet or savory variation of the pie. It may be that offering just one kind of snack made people feel limited in their choice [61,62]. Generally, people are attached to choice and feel curtailed when there are few options [63]. If it is true that offering only one type of snack frustrates the need for choice, it would be advisable not to offer the snack in a special designed food truck but instead to include it among a more regular range of snack outlets, in this case, for example, within the station, as proposed above. Alternatively, a future study could also just change one snack parameter such as assessing how a novel healthy snack compares to a similar novel unhealthy snack in an outlet. This to examine whether people would actually choose a healthy snack over an unhealthy one.

Third, the GPS study in Phase 2 of this project revealed that the majority of people snack at home, which is consistent with findings of earlier studies, showing that most of people’s snacking occurs at home [31,32]. However, out-of-home hotspots for unhealthy snacking with few healthy snack alternatives make it especially difficult for people to stick to a healthy diet since such environments offer almost no escape from eating unhealthy snacks [36]. Therefore, in this project, we focused on offering a healthier alternative at a hotspot for unhealthy snacking “on the go”. With the abovementioned explanations for our low sales, we hope to inspire future research to further target these unhealthy snacking hotspots.

Fourth and last, the included samples in the present studies were, on average, younger adults (M = 31 years; SD = 14). Future studies should aim to include participants from a broader age range to generalize our findings among all age groups.

## 7. Conclusions

Obesity and associated public health issues are a global problem and snacking plays an important role in this issue [64]. Changing people’s snack choices by offering healthy snacks at out-of-home hotspots for unhealthy snacking is thus a promising strategy for improving people’s diets. Our urban snack project was a pioneering effort to apply this approach and mainly demonstrated the complexity of such interventions in real life. Nevertheless, by exploring potential explanations for our findings, we hope to have provided new directions for future research on how a healthy and attractive snack can best be brought to the attention of consumers in order to give them the opportunity to buy a healthy snack in accordance with their stated intention to snack more healthily “on the go”.

## Figures and Tables

**Figure 1 ijerph-17-07538-f001:**
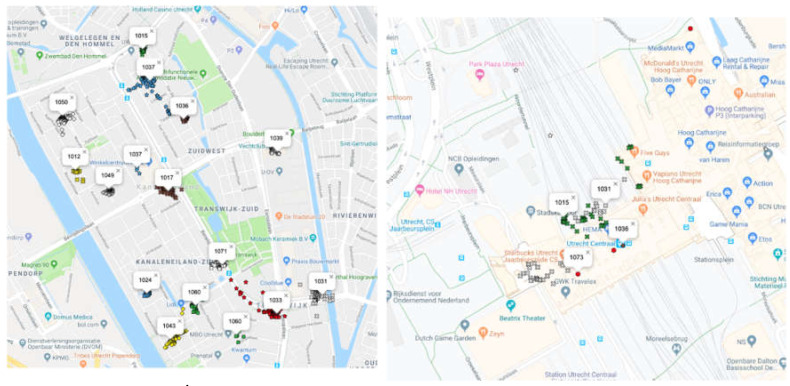
The map on the left shows individual survey participants who consumed snacks at home. The map on the right shows the GPS cluster in a 300 m radius centered on Utrecht Central Station, which includes at least four unique participants.

**Figure 2 ijerph-17-07538-f002:**
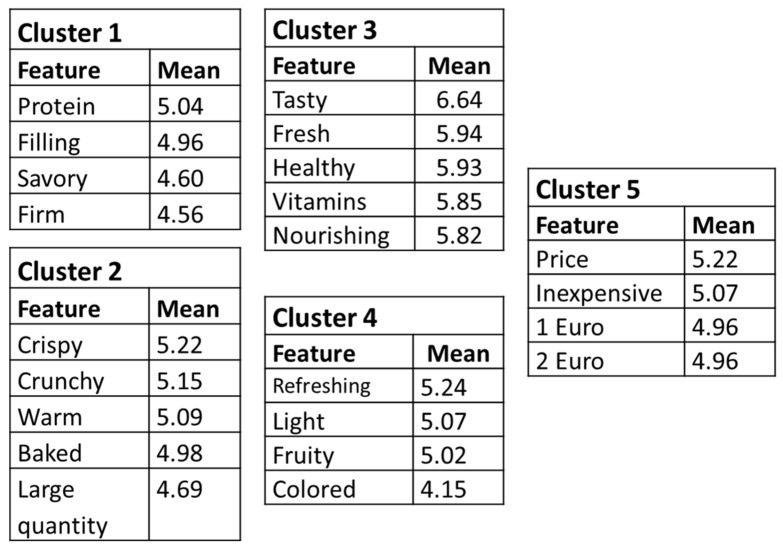
Concept-mapping clusters and feature ratings of the snack features (72 cards). Mean ratings indicate the importance given to a feature, while clustering indicates which features were felt to belong together.

**Figure 3 ijerph-17-07538-f003:**
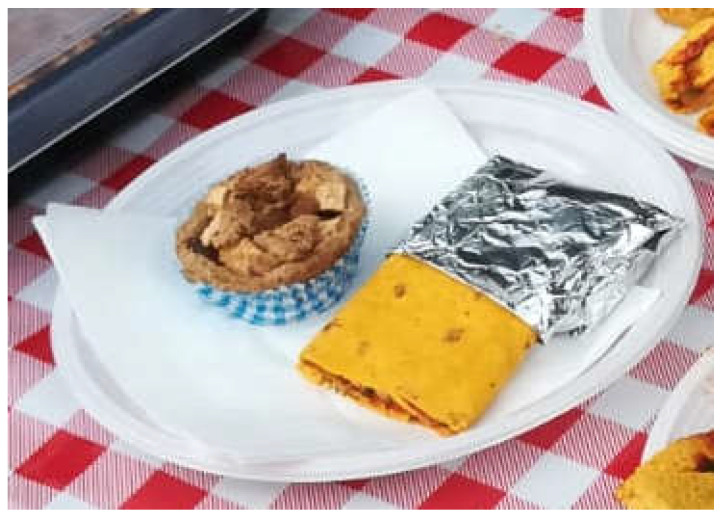
Examples of the pilot snacks: apple pie (left) and vegetable wrap (right).

**Figure 4 ijerph-17-07538-f004:**
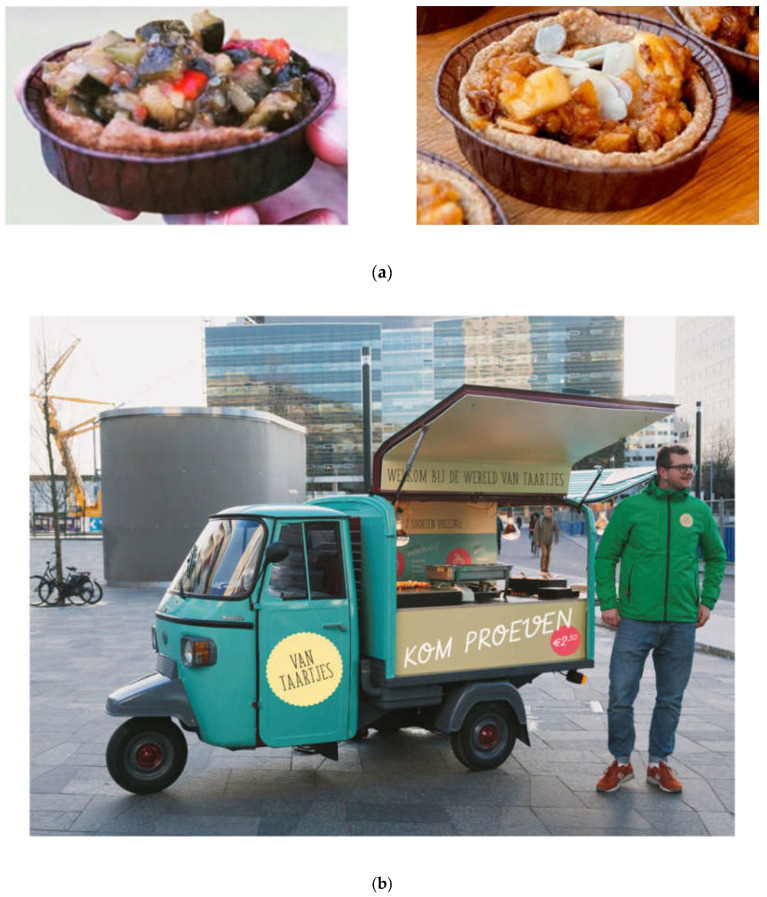
(**a**). Ratatouille pie (savory) on the left and apple and raisin pie (sweet) on the right. (**b**) Design of the mobile food car. The person depicted agreed to be shown. (**c**) Menu design (in Dutch). The menu describes the two snack offers and their filling ingredients: ratatouille pie (pepper, zucchini, eggplant, and tomato); apple and raisin pie (apple, cinnamon, and raisins).

**Figure 5 ijerph-17-07538-f005:**
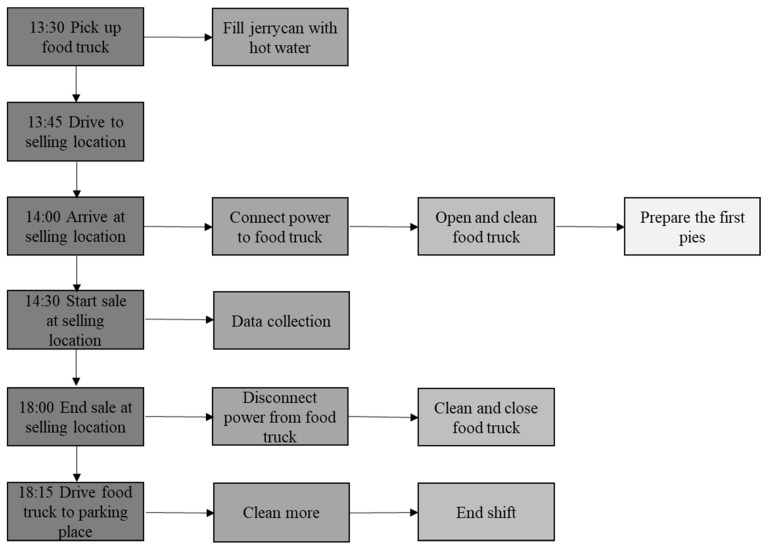
Flowchart of the field intervention procedures.

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
