# Peer review of "Snacks and The City: Unexpected Low Sales of an Easy-Access, Tasty, and Healthy Snack at an Urban Snacking Hotspot"

_ijerph, 2020, doi:10.3390/ijerph17207538_

Round 1

Reviewer 1 Report

The title can be changed if the study was mainly conducted around urban train station.

Since the average age of samples is 30 years old, I suggest that the conclusion has to be supported by this sampling method.

No other comments.

Author Response

Reviewer 1 (R1)

  1. The title can be changed if the study was mainly conducted around urban train station.

To make the title better fitting to the study objectives, we changed the title to: Snacks and the City: Unexpected low sales of an easy-access, tasty, and healthy snack at an urban snacking hotspot

  1. Since the average age of samples is 30 years old, I suggest that the conclusion has to be supported by this sampling method.

To address this point, we added this limitation to the Discussion section as follows (line 528): “Fourth and last, the included samples in the present studies were on average younger adults (M = 31 years; SD = 14). Future studies should aim to include participants from a broader age range to generalize our findings among all age groups. ”

Reviewer 2 Report

Comments:

This is an interesting study assessing if the accessibility of healthy snacks leads to healthier snacking. The comments below are restricted to those which reflect suggested changes for improvement.

Abstract
Line 27 Instead of saying “In the Discussion section we offer a number of explanations for these low sales, as well as suggestions for future research.”- provide a concise summary of the explanations in a sentence. The abstract should prompt readers to read the full text yet still summarise adequately the major findings and interpret them concisely.

Introduction

It may add value to refer to the Ottawa Charter for Health Promtion (1986):

https://www.who.int/healthpromotion/conferences/previous/ottawa/en/

In your case you are addressing the “enable” part by increasing the availability of healthier choices.

E.g. in Line 92 you could refer to this as the enablement is a proposed effective way to intervene.

Line 115 change “overweightness” to “overweight” – it is both an adjective and a noun

Lines 145- 149 Suggest rewriting your objectives in a clear manner following PICOTS.

Lines 149-168 This section introduces methods and it is best suited in the methods section.

Methods and Results

Line 174 list the Ethics approval number

Line 174 potential source of selection bias

Line 183 yes but was your intention to target this group? This is not clear in your objective. See comments above. In your objective participants are defined as “people”, rewrite your objective using PICOTS and if you wish to focus on lower SES state this clearly.

Line 195 define “app” before you use the shortened version

Line 199 eating snacks were ultimately analysed

Line 200 when a GPS

Line 202 GPS data were stored

Line 213 were present

Line 213 GPS data were condensed

Line 249 “The findings of Phase 1 can be summed up as follows.” Is not needed

Line 250 Here you are discussing the finding. Introducing context and citing previous research and comparing your findings to that belongs in the discussion section.

Line 253 again this belongs in the discussion section.

Line 293 How are “Vitamins”, “Healthy”, “Protein”, “Nourishing” sensory characteristics?

Lines 295-297 again this belongs in the discussion section.

Lines 369-371 you only need the information re average price of other snacks once, remove the second brackets.

Line 390 price reduction. (fullstop – get rid of rest – avoid repeating info)

Lince 397 marketing and branding is a science and art – was an expert consulted to provide marketing/branding insight and assist with this?

Figure 2

What is the difference between “Fresh” and “Garden-Fresh”?

How do you interpret the findings from Cluster 5? What does “price” mean compared to “inexpensive” as an answer?

Figure 5

14:30 is mentioned twice – too much redundancy!

Discussion

Line 446 you state “Future directions” but instead of this you provide limitations here. Limitations are also not well informed

My understanding is that no expert was involved to drive the marketing strategy of this new product and thus “prominently marketed” should be replaced with “prominently situated”. Marketing is a science and art that requires extensive experience for successful implementation. Big companies spend as much if not more budget on this than the actual product development and operations budgets. This is perhaps the biggest limitation of your study. You do not know if the result was due to people not wanting to but the healthy product, or if it is due to the established purchasing culture. It is hard for a new product to penetrate an established market and generate sales. And I would argue impossible without the relevant spending. You were competing with established brands and products that have vast spendings to drive their sales. If you offered a healthy and an unhealthy snack and the people chose the unhealthy one then yes you can make your claims. But if you only offer a novel unknown to the customer healthy snack and compete with established brands in the market this is a different story.

As such a future experiment could involve what proposed. Ie test the sale of novel unhealthy snack vs a novel healthy snack, this way you only change one parameter and you can assess the choice between healthy and unhealthy. Alternatively you can assess the sale of an established healthy vs an established unhealthy snack, same premise of changing one parameter rather than two at a time and not knowing which of the two informed the final decision.

As mentioned before you can cite the Ottawa Charter for Health Promtion (1986):

https://www.who.int/healthpromotion/conferences/previous/ottawa/en/

Another reason why your experiment did not work, may be that although the “enable” part was addressed the “advocate” and “mediate” part were not. Thus it is important to educate public and involve other sectors to ensure success of health behaviour changes. Future research can also test this.

General comment

There is a lot of repetition/redundancy. I would consider removing repeating chunks of text for brevity and to maintain the reader’s interest. I personally lose interest if I am encountering the same info over and over…

e.g. Lines 230-231 “According to the verbal reports of the people we asked in the  neighborhood, one important reason for snacking at home was that they considered out-of-home snacking to be expensive and thus preferred to buy snacks from local supermarkets to eat at home”

could be condensed to

“Participants reported cost of snacks informed choices.”

There are no clear borders between objectives/methods and Results/discussion. These specific comments above. Certain journals will allow you to combine methods/results or results/discussion sections but here there is an overlap all over the place.

Author Response

Reviewer 2 (R2)

This is an interesting study assessing if the accessibility of healthy snacks leads to healthier snacking. The comments below are restricted to those which reflect suggested changes for improvement.

We would like to thank R2 for the kind words.

Abstract
1. Line 27 Instead of saying “In the Discussion section we offer a number of explanations for these low sales, as well as suggestions for future research.”- provide a concise summary of the explanations in a sentence. The abstract should prompt readers to read the full text yet still summarise adequately the major findings and interpret them concisely.

We changed this sentence to (line 31): “In the Discussion, we name predominant eating and purchasing habits as possible reasons for these low sales. Future research could focus on placing the healthy snack directly beside people’s habitual snack purchase location to ensure that the new choice gets better recognized. “

Introduction

  1. It may add value to refer to the Ottawa Charter for Health Promtion (1986):

https://www.who.int/healthpromotion/conferences/previous/ottawa/en/

In your case you are addressing the “enable” part by increasing the availability of healthier choices.

E.g. in Line 92 you could refer to this as the enablement is a proposed effective way to intervene.

To address R2’s point to mention that our project relates to the World Health Organization health promotion goals, we added this reference to the introduction (line 77).

  1. Line 115 change “overweightness” to “overweight” – it is both an adjective and a noun

We have done so.

  1. Lines 145- 149 Suggest rewriting your objectives in a clear manner following PICOTS.

To address R2’s point, we now state our study objectives in the introduction more clearly by adding a separate subheading (e.g. starting line 151). Moreover, we made a separate subheading for the methods to clearly separate the Method and Results sections. However, the journal does not require a PICOT format for writing research papers and we also think that our research project does not meet the format of PICOT, as it was not a clinical trial implementing a treatment with a comparable reference group. (Taken from, https://www.ncbi.nlm.nih.gov/pmc/articles/PMC3430448/)

  1. Lines 149-168 This section introduces methods and it is best suited in the methods section.

We address this point in our answer to R2 in point 4.

 Methods and Results

  1. Line 174 list the Ethics approval number

We have now included the ethics approval number. (now line 169)

  1. Line 174 potential source of selection bias

To emphasize that we tried to avoid selection bias as much as possible, we now state that we randomly approached people (now line 166).

  1. Line 183 yes but was your intention to target this group? This is not clear in your objective. See comments above. In your objective participants are defined as “people”, rewrite your objective using PICOTS and if you wish to focus on lower SES state this clearly.

The Method section now includes the participant subsection in which we define our participant target group. We now clearly state that everyone from the city community between 16 years and older could participate and that we explicitly encouraged people from lower social economic backgrounds to take part in the studies.

  1. Line 195 define “app” before you use the shortened version

We now define app as mobile application (now line 227).

  1. Line 199 eating snacks were ultimately analysed

We changed the sentence: “Only the GPS data on the locations where people had recorded eating snacks was ultimately analysed.” to your suggestion above (now line 232).

  1. Line 200 when a GPS

We have done so. (now line 234)

  1. Line 202 GPS data were stored

We changed the sentence: “The GPS coordinates were automatically collected” to your suggestion above (now line 234).

  1. Line 213 were present

We changed the word “was” to “were” (now line 248).

  1. Line 213 GPS data were condensed

We changed the word “was” to “were” (now line 251).

  1. Line 249 “The findings of Phase 1 can be summed up as follows.” Is not needed

We removed this part (now line 287).

  1. Line 250 Here you are discussing the finding. Introducing context and citing previous research and comparing your findings to that belongs in the discussion section.
  2. Line 253 again this belongs in the discussion section.
  3. Lines 295-297 again this belongs in the discussion section.

Addressing R2’s point 16 to 18: To clarify that we summarize project phase 1 we now made clear subheadings.

  1. Line 293 How are “Vitamins”, “Healthy”, “Protein”, “Nourishing” sensory characteristics?

We agree with R2 that our cluster definitions were not yet clearly stated. We therefore added that participants named sensory and health-related characteristics, to make clear that words related to sensory characteristics as well as to the topic of health (now line 334).

  1. Lines 369-371 you only need the information re average price of other snacks once, remove the second brackets.

We have done so. (now line 416).

  1. Line 390 price reduction. (fullstop – get rid of rest – avoid repeating info)

We removed this. (now line 435).

  1. Line 397 marketing and branding is a science and art – was an expert consulted to provide marketing/branding insight and assist with this?

This is a good point of R2. We now more explicitly mention that the “food designers” came from a leading marketing company in the Netherlands that is specialized in marketing healthy food (now line 305).

 Figure 2

  1. What is the difference between “Fresh” and “Garden-Fresh”?

In the Dutch language there are two different words for fresh. To make clear that people used both words, but that they are not the same words, we now changed the translations to “fresh” and “refreshing”.

  1. How do you interpret the findings from Cluster 5? What does “price” mean compared to “inexpensive” as an answer?

We only interpreted the data on a cluster level and not the individual word level. All words named in cluster 5 belong to the semantic idea of price, i.e., 1 euro, 2 euro, inexpensive, price. We therefore concluded that price would be an important factor for people in their ideal snack.

 Figure 5

  1. 14:30 is mentioned twice – too much redundancy!

Thank you for catching this error. We removed one of them.

 Discussion

  1. Line 446 you state “Future directions” but instead of this you provide limitations here. Limitations are also not well informed

To make sure that it is clear that the section discusses the study limitations and provides future directions for offering healthy snacks, we changed this subheading to “Study limitations and future directions for offering easy-to-access healthy snacks“ (now line 491).

  1. My understanding is that no expert was involved to drive the marketing strategy of this new product and thus “prominently marketed” should be replaced with “prominently situated”. Marketing is a science and art that requires extensive experience for successful implementation. Big companies spend as much if not more budget on this than the actual product development and operations budgets. This is perhaps the biggest limitation of your study. You do not know if the result was due to people not wanting to but the healthy product, or if it is due to the established purchasing culture. It is hard for a new product to penetrate an established market and generate sales. And I would argue impossible without the relevant spending. You were competing with established brands and products that have vast spendings to drive their sales. If you offered a healthy and an unhealthy snack and the people chose the unhealthy one then yes you can make your claims. But if you only offer a novel unknown to the customer healthy snack and compete with established brands in the market this is a different story.

See also our answer to point 22 to R2. R2’s concern arises from the fact that we did not adequately state that we actually worked together with a leading food marketing company that helped us during the entire process of developing and branding the snack. In other words, the whole development and marketing strategy was made in close collaboration with experts in the marketing field, www.foodcabinet.nl. For the interested reader, we also included their website in the manuscript.

  1. As such a future experiment could involve what proposed. Ie test the sale of novel unhealthy snack vs a novel healthy snack, this way you only change one parameter and you can assess the choice between healthy and unhealthy. Alternatively you can assess the sale of an established healthy vs an established unhealthy snack, same premise of changing one parameter rather than two at a time and not knowing which of the two informed the final decision.

This is indeed a good point. We therefore added R2’s suggestion to the Discussion as a possible route for future research as follows (line 516): ”Alternatively, a future study could also just change one snack parameter such as assessing how a novel healthy snack compares to a similar novel unhealthy snack in an (existing) outlet. This would allow to examine whether people would actually choose for a healthy snack over an unhealthy one. ”  

  1. As mentioned before you can cite the Ottawa Charter for Health Promtion (1986):

https://www.who.int/healthpromotion/conferences/previous/ottawa/en/

Another reason why your experiment did not work, may be that although the “enable” part was addressed the “advocate” and “mediate” part were not. Thus it is important to educate public and involve other sectors to ensure success of health behaviour changes. Future research can also test this.

See also our answer in point 2 to R2.

Moreover, our project involved parties of different stakeholders such as the local government and leading food marketeers, as well as experts from different research areas such as geography, medicine and public health. To make this clear, we now added this information at line 159. Additionally, we already mentioned in the discussion that peoples existing snacks habits might have been too strong and that our environmental intervention was too small to change these strong snacking habits. However, to also make a suggestion for future research to this point, we added to the discussion that a future study could involve an environmental prompt in combination with a clearly stated educational prompt to also remind people on their healthy eating goal (line 505): “Another option could be to more prominently promote the snack as healthy in form of a health campaign, maybe even in collaboration with other parties such as the local government or popular snack outlets. This to bundle energies by not only offering a new snack, but also educating people about healthy snack options.”.

General comment

  1. There is a lot of repetition/redundancy. I would consider removing repeating chunks of text for brevity and to maintain the reader’s interest. I personally lose interest if I am encountering the same info over and over…

e.g. Lines 230-231 “According to the verbal reports of the people we asked in the  neighborhood, one important reason for snacking at home was that they considered out-of-home snacking to be expensive and thus preferred to buy snacks from local supermarkets to eat at home”

could be condensed to

“Participants reported cost of snacks informed choices.”

To address R2’s point, we shortened all text that R2 requested us to do. Additionally, we reread the entire article and shortened the text where we thought this would be possible. Moreover, we had our manuscript professionally edited by a native English speaker.

  1. There are no clear borders between objectives/methods and Results/discussion. These specific comments above. Certain journals will allow you to combine methods/results or results/discussion sections but here there is an overlap all over the place.

Please see our answer in point 4 to R2.

Reviewer 3 Report

The manuscript is very interesting and novel. The authors evaluated a very important aspect regarding "Snacks". Are snachs healthy or not ?, and people are arguing to change their behavior in front of food ?. This type of study is very important in food, nutrition and public health. Especially in those countries where overweight and obesity has had a great growth.

I only have some comments that I think could improve the quality of the manuscript.

I. Major comment:
1. In the introduction I suggest including a brief paragraph regarding the components of the diet (saturated fat, refined carbohydrates, syrups rich in fructose) that are associated with an increased risk of developing obesity, CVD, NAFLD, etc.
2. It would be interesting to briefly discuss how specific components of snacks (saturated fat, refined carbohydrates "starches and sugar", etc.) increase the risk of obesity and other chronic diseases in the population.

Suggested references:

The Impact of Maternal Diet during Pregnancy and Lactation on the Fatty Acid Composition of Erythrocytes and Breast Milk of Chilean Women. Nutrients. 2018; 10: 839. PMID: 29958393

Mozaffarian et al. Changes in diet and lifestyle and long-term weight gain in women and men. N Engl J Med. 2011; 364: 2392-404.

II. Minor comment:
1. Improve the writing of the study objective.

Author Response

WE ALSO INCLUDED THE ENTIRE COVER LETTER AS AN ATTACHEMNT BELOW

Reviewer 3 (R3)

The manuscript is very interesting and novel. The authors evaluated a very important aspect regarding "Snacks". Are snachs healthy or not ?, and people are arguing to change their behavior in front of food ?. This type of study is very important in food, nutrition and public health. Especially in those countries where overweight and obesity has had a great growth.

We would like to thank R3 for the kind words.

  1. Major comment:
  2. In the introduction I suggest including a brief paragraph regarding the components of the diet (saturated fat, refined carbohydrates, syrups rich in fructose) that are associated with an increased risk of developing obesity, CVD, NAFLD, etc.

The study objectives are about the potential of rearranging the obesogenic environment and not about specific diet components that relate to obesity. Moreover, in the introduction, we already state that (line 42 and onwards) “Recent estimates indicate that some 25–35% of daily energy intake now comes from snacking (Piernas & Popkin, 2010a; Piernas & Popkin, 2010b). This is problematic because snacking is strongly associated with unhealthy diets (Bertheus-Forsland, Torgerson, Sjostrom, & Lindroos, 2005; Howarth, Huang, Roberts, Lin, & McCrory, 2007), mainly because snacking tends to involve the consumption of energy-dense foods that are rich in sugar and/or saturated fat and of poor nutritional quality.”

  1. It would be interesting to briefly discuss how specific components of snacks (saturated fat, refined carbohydrates "starches and sugar", etc.) increase the risk of obesity and other chronic diseases in the population.

Suggested references:

The Impact of Maternal Diet during Pregnancy and Lactation on the Fatty Acid Composition of Erythrocytes and Breast Milk of Chilean Women. Nutrients. 2018; 10: 839. PMID: 29958393

Mozaffarian et al. Changes in diet and lifestyle and long-term weight gain in women and men. N Engl J Med. 2011; 364: 2392-404.

Please see our answer to point 1 to R3.

Moreover, we state in the conclusion section that (line 534): “Obesity and associated public health issues are a global problem, and snacking plays an important role in this issue (Heymsfield & Wadden, 2017; Nielsen et al., 2002). Changing people’s snack choices by offering healthy snacks at out-of-home hotspots for unhealthy snacking is thus a promising strategy for improving people’s diets.”

  1. Minor comment:
    1. Improve the writing of the study objective.

Please see our answer in point 4 to R2.

Round 2

Reviewer 2 Report

Thank you for attending to the comments. I hope the feedback was useful for the improvement of the article.